# Influence of Lipoxygenase Inhibition on Glioblastoma Cell Biology

**DOI:** 10.3390/ijms21218395

**Published:** 2020-11-09

**Authors:** Felipe da Costa Souza, Matthew Thomas Ferreira, Alison Colquhoun

**Affiliations:** Department of Cell and Developmental Biology, Institute of Biomedical Sciences, University of São Paulo, São Paulo 03001-000, Brazil; matthewferreira@usp.br (M.T.F.); alison@usp.br (A.C.)

**Keywords:** GBM, OXLAM, lipoxygenase, glioblastoma, luteolin, hydroxyoctadecadienoic acid, hydroxyeicosapentaenoic acid, linoleic acid, 13-HODE, eicosanoids

## Abstract

Background: The relationship between glioblastoma (GBM) and fatty acid metabolism could be the key to elucidate more effective therapeutic targets. 15-lipoxygenase-1 (15-LOX), a linolenic acid and arachidonic acid metabolizing enzyme, induces both pro- and antitumorigenic effects in different cancer types. Its role in glioma activity has not yet been clearly described. The objective of this study was to identify the influence of 15-LOX and its metabolites on glioblastoma cell activity. Methods: GBM cell lines were examined using high-performance liquid chromatography-tandem mass spectrometry (HPLC-MS/MS) to identify 15-LOX metabolites. GBM cells treated with 15-LOX metabolites, 13-hydroxyoctadecadeinoic acid (HODE) and 9-HODE, and two 15-LOX inhibitors (luteolin and nordihydroguaiaretic acid) were also examined. Dose response/viability curves, RT-PCRs, flow cytometry, migration assays, and zymograms were performed to analyze GBM growth, migration, and invasion. Results: Higher quantities of 13-HODE were observed in five GBM cell lines compared to other lipids analyzed. Both 13-HODE and 9-HODE increased cell count in U87MG. 15-LOX inhibition decreased migration and increased cell cycle arrest in the G2/M phase. Conclusion: 15-LOX and its linoleic acid (LA)-derived metabolites exercise a protumorigenic influence on GBM cells in vitro. Elevated endogenous levels of 13-HODE called attention to the relationship between linoleic acid metabolism and GBM cell activity.

## 1. Introduction

Glioblastoma (GBM) is a grade IV astrocyte-derived tumor and the most aggressive type of glioma, corresponding to 65% of identified gliomas [1]. Despite the lower incidence compared with other human tumors, GBM is characterized by a high mortality, with a median survival of 12–15 months following standard treatment [2,3,4]. As an extremely aggressive tumor, GBM has been extensively studied to identify new molecular targets that could lead to new strategies and adjuvant therapies, improving the results of the standard treatments.

As reviewed by Hanahan and Weinberg in 2011, inflammation is an emerging cancer hallmark, intimately connected with microenvironmental changes responsible for promoting other cancer hallmarks that increase tumor development [5]. The roles of n-3 and n-6 polyunsaturated fatty acid (PUFA) metabolisms upon cell signaling, inflammation, metabolism, and proliferation have been extensively studied over the past years [6,7]. Despite their known biological functions, PUFAs are also substrates for the synthesis of lipid-derived molecules collectively called oxylipins. PUFAs, metabolized by cyclooxygenase (COX), lipoxygenases (LOX), and/or cytochrome P450 enzymes, produce molecules that can act as signals in a range of physiological and pathological processes, including cancer [8,9,10].

Depending on whether the substrate is n-6 or n-3 PUFA, these pathways can produce an array of molecules. The lipoxygenases oxidize fatty acids such as n-6 arachidonic acid (AA) and n-6 linoleic Acid (LA) (Figure 1). Human lipoxygenases are 5-lipoxygenase (5-LOX), 12-lipoxygenase (12-LOX, also called platelet-type 12-lipoxygenase), and 15-lipoxygenase (15-LOX, subtypes -1 and -2) [11]. The 5-LOX enzyme is the most well-characterized, catalyzing the conversion of AA into 5-hydroperoxyeicosatetraenoic acid (5-HpETE), an unstable and inactive precursor of 5-hydroxyeicosatetraenoic acid (5-HETE) and leukotriene A_4_ (LTA_4_). LTA_4_ is also a precursor of biologically active leukotrienes, which depend on LTA_4_ hydrolase (LTA4H) and LTC_4_ synthase (LTC_4_S) activity to produce LTB_4_ and LTC_4_, respectively. LTC_4_, in turn, can be converted to LTD_4_ and LTE_4_ [12]. Additionally, 5-LOX activities are dependent on a co-factor: 5-lipoxygenase-activating protein (FLAP), which is required for 5-LOX binding with AA [13].

The 12-lipoxygenase pathway catalyzes AA conversion to 12-hydroperoxyeicosatetraenoic acid (12-HpETE), also an unstable and inactive precursor to 12-hydroxyeicosatetraenoic acid (12-HETE) formation. 12-HETE’s biological activities are associated with vessel permeability, vasoconstriction, vasodilation, and platelet aggregation during inflammation and tissue repair [14,15,16,17,18].

Fatty acid metabolism through 15-LOX occurs by two functional isomeric enzymes designated as 15-LOX-1 and 15-LOX-2. Both can catalyze the conversion of AA into 15-hydroperoxyeicosatetraenoic acid (15-HpETE) and into the functional 15-hydroxyeicosatetraenoic acid (15-HETE), a potent proangiogenic signaling molecule [19,20]. Despite 15-LOX-1′s ability to interact with AA, it has a greater affinity toward LA, making LA its main substrate [21]. 15-LOX-1 converts LA into the 18-carbon OXLAMs (oxidized linoleic acid metabolites) 13-hydroxyoctadecadienoic acid (13-HODE) and 9-hydroxyoctadecadeinoic acid (9-HODE) [22]. Most of 15-HETE’s and HODEs’ activities are triggered through the activation of peroxisome-proliferator-activated receptor (PPAR) gamma and PPAR beta/delta, as well as G-protein coupled receptor 132 (GPR132), which binds with 9-HODE [23]. Through receptor activation, these 15-LOX metabolites influence a wide range of cellular activities, from growth to migration and invasion. In addition to AA and LA, 15-LOX-1 can also metabolize n-3 docosahexaenoic acid (DHA), producing 17-hydroxy docosahexaenoic acid (17-HDHA), a docosanoid precursor of the anti-inflammatory resolvins [24].

The relationship between the lipoxygenase pathways and cancer is less characterized than the cyclooxygenase pathway. The expression of 5-LOX and 12-LOX are normally low or absent in normal tissues, being expressed only under pathological conditions like cancer. The upregulated expression of 5-LOX and 12-LOX have been identified in colon, prostate, breast, and pancreas cancers and associated with tumor cell proliferation, migration, and worsened prognosis [25,26,27,28,29]. Interestingly, the expression of the 15-LOXs (isoforms 1 and 2) vary between different cancer types and must be further examined.

These interesting findings raise more questions about LOX-derived oxylipins and cancer. However, the role of each oxylipin in GBM has not been widely investigated, especially regarding lipoxygenases and cytochrome P450 pathways. In this study, we aimed to identify the LOX and cytochrome P450 (CYP450)-related enzymes and receptors expressed in GBM cell lines, along with the (S)-form oxylipin they produced. To elucidate the specific roles of the more prominent products, we exogenously treated GBM cells with oxylipins and pharmacological inhibitors of the LOX pathway (Figure 1), followed by a series of functional assays to assess cell proliferation, death, migration, and invasive capacity.

## 2. Results

### 2.1. qRT-PCR and RT-PCR Transcriptional Prolifes of Genes Involving the Lipoxygenase and Cytochrome P450 Pathway

To evaluate the importance of the lipoxygenase and cytochrome P450 components in GBM, the mRNA levels of the following enzymes involved in the metabolism of fatty acids were examined: 5-LOX, LTA4H, LTC4S, FLAP, 12-LOX, CYP4A11 (Cytochrome P450 Family 4 Subfamily A Member 11), 15-LOX-1, and 15-LOX-2. The receptors CYSLTR2 (Cysteinyl Leukotriene Receptor 2), BLT1 (Leukotriene B4 receptor 1), BLT2 (Leukotriene B4 receptor 2), PPAR alpha, PPAR beta/delta, PPAR gamma, and GPR132 in all cell lines were also examined. The results showed that all genes analyzed were expressed heterogeneously in all the cell lines examined (Figure 2A). 5-LOX mRNA was found in A172 but not in the other cell lines. Surprisingly, FLAP mRNA was found at higher levels in A172, U251-MG, and U87-MG, regardless of 5-LOX expression. In fact, A172 had the lowest level of FLAP mRNA compared with U251-MG and U87-MG. The same was seen with the mRNA of leukotriene-producing enzymes LTA4H and LTC4S, found at different levels independent of FLAP or 5-LOX expression. U87-MG demonstrated low levels of both LTB_4_ receptors, BLT1 and BLT2. In fact, U87-MG had an overall lower expression of all the genes, except for FLAP, which had a more pronounced expression than the other cell lines. All five cell lines expressed mRNA for the 15-LOX pathway, as observed using conventional PCR (Figure 2B). PPAR expression was heterogenous among the cell lines. A172 and U87-MG showed reduced PPAR beta/delta expression but showed higher PPAR gamma expression compared with U251-MG, U138-MG, and T98G. PPAR alpha expression was higher in T98G (Figure 2A).

### 2.2. Oxylipin Production in U251-MG, U87-MG, U138-MG, T98G, and A172 Cells

Following the mRNA profile, the oxylipin production profile was evaluated in GBM cell lines using liquid chromatography-electrospray ionization-tandem mass spectrometry (LC-ESI-MS/MS). The following panel of three (S)-form eicosanoids, one docosanoid, and two OXLAMs, derived from fatty acid metabolism by the lipoxygenase pathway were quantified: 13-HODE, 9-HODE, 15-HETE, 17-HDHA, 12-HETE, and 5-HETE. Levels of 13-HODE were similar in all cell lines analyzed (Table 1). Levels of 9-HODE were also detected in U251-MG, U138-MG, and T98G. None of the leukotrienes from the 5-LOX pathway were detected among any of cell line products. A low concentration of 5-HETE was found in A172, the only cell line that expressed 5-LOX mRNA by real-time PCR. Regarding 12- and 15-HETE production, U87-MG and U251-MG were the only cell lines to produce 12-HETE and T98G the only cell line to produce 15-HETE. Levels of 17-HDHA were found in low concentrations in T98G and higher in A172. Thus, based on the mRNA and LC-MS/MS profile, three cell lines were initally chosen to represent GBM cells with 15LOX/13-HODE profiles: U251-MG, U87-MG, and A172. However, as the study progressed, we identified T98G as another cell line of interest due to its detectable production of both 13-HODE and 15-HETE, as well as its history of being a more drug-resistant cell line. As summarized in Table 1, U251-MG produced fewer 5-LOX products, more 12-LOX mRNA, and more 12-HETE; U87-MG produced fewer 5-LOX products, less 12-LOX mRNA, and more 12-HETE; and A172 produced more 5-LOX products, less 12-LOX mRNA, and no detectable 12-HETE.

### 2.3. Influence of Treatments on Cell Viability (3-(4,5-Dimethylthiazol-2-yl)-2,5-Diphenyltetrazolium Bromide (MTT)) and Cell Count

The next step was to evaluate the cell viability through MTT assays following the exogenous administration of 13-HODE, 9-HODE, 15-HETE, 5-HETE, 12-HETE, 8-HETE, 9-HETE, and 20-HETE. Reduced viability was observed in the U251-MG cell line after 0.5 µM and 1 µM 13-HODE, 9-HODE, 15-HETE, 5-HETE, and 8-HETE treatments at 48 h (Appendix A). After 72 h, 13-HODE, 9-HODE, and 15-HETE treatments reduced U251-MG cell viability significantly but only with 0.1 µM and 0.5 µM concentrations. 5-HETE also reduced U251-MG viability at 72 h but with all concentrations (Figure 3A). The U87-MG and A172 cell lines showed no statistical differences between treatments and their respective controls, regardless of the time and concentration (Figure 3B,C and Appendix A). To determine if greater concentrations of 13-HODE and 9-HODE would illicit any responses, we treated U87-MG and T98G with either 9-HODE or 13-HODE at 1 µM, 5 µM, and 10 µM for 24, 48, and 72 h. Interestingly, U87-MG increased in cell number with 5 µM of 13-HODE and both 5 and 10 µM of 9-HODE at 72 h. T98G did not demonstrate any significant response (Figure 3D,E).

### 2.4. The 15-Lipoxygenase Inhibitors but Not 5-LOX Inhibitors Reduced GBM Cell Counts

After establishing the enzyme, receptor, and product profiles, as well as the effects of the oxylipins on cell growth, the modulation of lipoxygenase activities was explored using different pharmacological lipoxygenase inhibitors. Luteolin was used to inhibit 15-lipoxygenase-1, while NDGA, considered a pan LOX-inhibitor [30], was used to inhibit 12-LOX/15-LOX-2 [31]. CAY10606 and CAY10649 were used as direct inhibitors of 5-lipoxygenase, while MK886 disrupted 5-LOX activity indirectly by binding to FLAP. First, U251-MG, U87-MG, and A172 cells were treated for 72 h with different concentrations of luteolin, NDGA, CAY10606, CAY10649, and MK886 (Appendix A). A significant reduction in cell counts was seen in U87-MG and U251-MG after 72 h of 15-LOX inhibition (luteolin (15 µM)/NDGA (40 µM)) (Figure 4A). Phase-contrast microscope images also showed a decrease in cell confluence at 72 h (Figure 4D). The number of U87-MG cells also significantly decreased in response to one of the 5-LOX inhibitors (CAY10606). Since the three cell lines did respond to 15-LOX inhibition but did not produce detectable levels of 15-HETE at baseline, treatments applied to T98G, which produced detectable 15-HETE, were investigated. Both luteolin and NDGA also reduced the T98G cell count with 72 h of treatment (Figure 4E). U87-MG cells concomitantly treated with luteolin, and 13-HODE demonstrated an increase in cell count (Appendix A).

### 2.5. The 15-Lipoxygenase Inhibition Influenced Cell Cycle and Apoptosis in U87-MG

To complement the data concerning cell growth and proliferation as measured by the total number of cells, a cell cycle analysis with propidium iodide staining measured by flow cytometry after 72 h of treating U87-MG, U251-MG, and A172 was performed. The data indicate event distributions among the cell cycle phases through propidium iodide fluorescence quantification. There was no change in cell cycle distribution in U251-MG cells treated with luteolin or NDGA (Figure 5A). Moreover, only a small reduction in G2/M cells was seen in A172 with NDGA, while luteolin led to no changes in any phase (Figure 5C). The U87-MG cell line, on the other hand, had a reduction in the G1 phase and a concomitant increase in G2/M with both luteolin and NDGA compared with the control, suggesting an arrest of U87-MG cells at the G2/M checkpoint (Figure 5B). In parallel with the cell cycle analysis, analyses of the cell death induced after luteolin and NDGA treatments were carried out. The U251-MG, U87-MG, and A172 cells were treated for 72 h and incubated with annexin V and propidium iodide for further flow cytometry analysis. The results showed an increase in apoptotic U251-MG and U87-MG cells with luteolin but not with NDGA (Figure 5D–G). Neither luteolin nor NDGA were able to induce significant changes in the A172 cell line.

### 2.6. The 15-Lipoxygenase Inhibition Reduced the Migration of GBM Cells

Considering the previous results related to cell growth/cycle and using the previously chosen concentrations, Transwell assays to evaluate the influence of luteolin and NDGA treatments upon the in vitro migration of U251-MG, U87-MG, and A172 cells were performed. Cells migrated for 24 h while submitted to their respective treatments before staining and quantification. The results showed that both luteolin and NDGA reduced cell migration through the Transwell membrane. U251-MG migration was significantly reduced by 69% and 67% using Luteolin and NDGA, respectively, when compared with their dimethyl sulfoxide (DMSO) controls (Figure 6A). Similarly, U87-MG migration was reduced by 47% with NDGA, while luteolin’s 35% reduction was not statistically significant (Figure 6B). A172 migration was reduced by 37% and 49% using luteolin and NDGA, respectively; both were statistically significant (Figure 6C). The photomicrograph of Transwell membranes show representative images for each treatment and the DMSO control (Figure 6D). Additionally, a wound-healing assay conducted with T98G cells showed a reduction in cell migration with only 12 h of luteolin or NDGA treatments (Figure 6E and Appendix A). The migration of cells treated with 9-HODE and 15-HETE was not affected. However, 13-HODE did reduce T98G migration.

### 2.7. The 15-Lipoxygenase Reduced the Metalloprotease (MMP) Activity of GBM Cells

Luteolin and NDGA not only reduced the cell count over 24 and 72 h of treatment, but they also reduced the MMP-2 activity, thus limiting the invasive potential of the cells. Quantitative RT-PCR demonstrated the presence of MMP-2 mRNA and negligibly small amounts of MMP-9 mRNA in all five cells lines (Figure 7A,B). Zymography assays of the baseline expression demonstrated that latent and active forms of MMP-2 were present in both U87-MG and T98G, and there was no observable MMP-9 activity (Figure 7C,E,F). Cells treated with 13-HODE and 9-HODE demonstrated an increased expression of MMP-2 mRNA (Figure 7D). Interestingly, U87-MG cells treated with luteolin or NDGA demonstrated reduced MMP-2 activity compared with their respective controls. T98G, however, demonstrated an increase in MMP2. (Figure 7G,H). In both cell lines treated with NDGA in serum-free medium for 72 h, a complete inhibition of MMP2 was observed (Figure 7H).

### 2.8. The 15-Lipoxygenase-1 Synthesis Is Not Influenced by the LOX Inhibitors

To address whether the results of the functional assays herein described are due to the inhibition of 15-LOX activity or due to the reduced synthesis of 15-LOX, a Western blot was performed (Figure 8). Samples of the protein lysates of cell lines U87-MG (Figure 8A) and T98G (Figure 8B) treated with luteolin or NDGA for 72 h were tested. No significant reduction in band intensity when normalized to the sample control (β-actin) was identified in either cell line (Figure 8C,D). The LOX inhibitors (luteolin and NDGA) inhibited 15-LOX-1 activity and not its synthesis.

## 3. Discussion

The metabolism of PUFAs through the activity of lipoxygenases and cyclooxygenases in cancer, as reviewed by Gomes et al. (2018), has been a subject of discussion in the literature over the last decades. The synthesis of eicosanoids and other oxylipins from the oxygenation of PUFAs is related to the regulation of processes, such as cell proliferation, cell adhesion, migration, angiogenesis, and vascular permeability [32]. All these cellular processes are involved in the inflammatory response and are also related to the development of tumors [5].

The LOX pathway (Figure 1) in tumors has received less attention than the COX pathway, particularly in the central nervous system. The first aim of this study was to characterize the transcriptional profile of several components pertaining to the lipoxygenase pathway, including enzymes, receptors, and metabolites. These data showed a heterogeneous profile among the five in vitro lines analyzed, especially in their transcriptional profiles. The more tumorigenic cell line, U87-MG, was selected, along with U251-MG and A172, to identify the possible common roles of 15-LOX among these heterogenous cell lines. The second aim of this study was to explore the influence of LOX inhibition on cell growth, migration, and invasive capacity.

### 3.1. The 5-Lipoxygenase in the GBM Cell Lines

5-LOX activity is intimately connected with the inflammatory process, and, in cancer, its role is context-dependent. According to Zhang et al. (2006), 5-LOX is normally produced by neurons and glial cells. The inhibition of 5-LOX is essential for a neuroprotective effect, and in the context of brain traumas, 5-LOX is often upregulated [33]. It is possible that this initial inflammatory response could illicit a protumorigenic environment for transformed astrocytes. The Human Protein Atlas (HPA; at www.proteinatlas.org) confirms our findings that the cell lines U87-MG, U251-MG, and U138-MG do not express 5-LOX mRNA. They also demonstrated that high-grade gliomas also do not express 5-LOX mRNA [34]. Interestingly, a previous study reported an overexpression of 5-LOX in A172, which is similar to the data in our study. In the same report 5-LOX was found in all GBM and astrocytoma tumor samples analyzed [35], which conflicts with our data and data from the HPA. One possible explanation of the presence of 5-LOX measured in the patient cells could be due to the heterogenous cell population of the tumor microenvironment.

5-LOX converts 17-hydroxy docosahexaenoic acid (17-HDHA) into special proresolving mediators (SPM) (ex., resolvins) [36]. 17-HDHA is synthesized by 15-LOX from docosahexaenoic acid (DHA), which is the most abundant fatty acid in the brain. The presence of 17-HDHA in A172 and T98G cells warrants further investigation of the potential for SPMs production in glioma cells. A possible motivation for a tumor’s downregulation of 5-LOX could be due to its role in recruiting tumor-associated macrophages [37]. Resident innate immune cells of the central nervous system are microglia cells. Zhang et al. (2014) demonstrated that 5-LOX is responsible for microglial cell activation [38]. Thus, its suppression, even temporarily, may be necessary for the tumor’s survival. However, further investigation is required, because an overexpression of 5-LOX in U87-MG has also been reported in the literature [39].

As previously mentioned, 5-LOX is responsible for producing not only various proinflammatory leukotrienes but, also, 5-HETE. Our data show that, despite 5-HETE production in only A172, there was no difference in cell growth after treating A172, U87-MG, and U251-MG with three different 5-LOX inhibitors. The only exception was U87-MG with CAY10606 at 72 h (Figure 4B). Since there was no detection of 5-LOX mRNA on U87-MG, and considering the lack of effect with the other two inhibitors, this effect is most likely an off-target effect. This indicates that 5-LOX alone plays no direct, autocrine role in GBM cell growth in vitro. The treatment with exogenous 5-HETE led to no change in cell viability, with the exception of U251-MG (Figure 3A); therefore, the possible interaction between GBM cells and 5-LOX products derived from the microenvironment should not be discarded. Massi et al. (2008) demonstrated that the in vitro inhibition of 5-LOX using MK886 increased the antitumoral effects of cannabidiol in U87-MG, while the use of MK886 alone resulted in effects similar to those in the present study at similar concentrations [40]. The importance of preventing compensatory 5-LOX activation as a response to other treatments should not be discarded either. In order to produce 5-HETE from AA, FLAP is required. Interestingly, the FLAP mRNA expression was observed in A172, U251-MG, and U87-MG, regardless of 5-LOX expression. This pattern was also confirmed in the HPA [34]. The high expression of FLAP is also significantly associated with a decreased survival in GBM patients [41]. Other associated functions of FLAP in GBM cells remain to be elucidated.

### 3.2. The 15-Lipoxygenase’s Influence on the Growth, Migration, and Invasive Capacity in GBM Cell Lines

The LC-MS/MS data showed that, among the main metabolites of the LOX pathway, the 15-LOX-1 product, 13-HODE, is produced in all the cell lines. The other 15-LOX-1 product, 9-HODE, was detected in U251-MG, T98G, and U138-MG. The AA-derived 15-HETE (through 15-LOX-2) was found only in T98G (Figure 2). Despite the possible nonenzymatic oxidation of LA [42], the varying amounts of 15-LOX products suggest that nonenzymatic synthesis is not the main producer of HODEs/15-HETE in these cells. Moreover, although the HPA reported no mRNA expression of 15-LOX-1 in U138-MG, U251-MG, and U87-MG, their immunostaining of GBM patient samples demonstrated a moderate expression of 15-LOX-1 [34]. The TCGA database showed that the expression of 15-LOX-1 and -2 does not directly alter survival of the patient [41]. The increase of 13-HODE in GBM cells was also described in the U87-MG cell line responding to hypoxic stress [43].

In the normal adult rat brain, both 13-HODE and 9-HODE are detected as the most predominant OXLAMs [44,45]. Post-natal day zero to one rat brains not only show 13-HODE and 9-HODE in high levels, but 50% of total oxylipin found in developing brains are OXLAMs, as compared with the 5–7% found in adult rat brains [44,46]. Our results demonstrate that 13-HODE was the most abundant among all the oxylipins herein analyzed (Table 1), which resembles 13-HODE in a developing brain.

During the past decade, brain tumor research has focused on understanding the mechanisms regulating the phenotypes of glioblastoma stem cells. These “stem-like cells” are a self-renewing multipotent population, which express the neural stem cell surface marker, CD133, first isolated from human glioblastomas [47,48,49,50]. Other embryonic stem cell factors such as OCT4, SOX2, MYC, and KLF4 are also found differentially expressed in glioblastoma and normal neural stem cells [51]. In fact, a recent work showing single-cell RNA-sequencing revealed the same neurodevelopmental cell hierarchy, conserved between both glioblastoma and normal developing human brain tissue [52]. All this information, along with our data presented herein, raises interesting questions about 13-HODE and 9-HODE as a conserved feature between brain tumors, “stemness”, and the developing brain.

### 3.3. Growth

To comprehend the roles of 13-HODE and 9-HODE in GBM, examining their associated pathways in other models of cancer may be helpful. The roles of 15-LOX-1 and -2 described in cancer are most frequently associated with 13-HODE production. The documented influence of 13-HODE in cancer illustrates some perplexing controversies. In some reports, 13-HODE production is associated with anti-tumoral effects by suppressing PPAR Beta/Delta [53,54]. Moreover, the loss of 15-LOX-1, and the consequent reduction in 13-HODE levels has been associated with tumor progression in lung [55], colorectal [56], pancreatic [57], and breast cancers [58]. However, this relationship seems to be inverted in prostate cancer, which is similar to the data obtained herein with GBM cells. The induced overexpression of 15-LOX-1 in normal prostate tissues induced neoplastic lesions [59,60,61]. Furthermore, PC-3 and LNCaP prostate cancer cell lines generate more 13-HODE, and treating PC-3 with exogenous 13-HODE led to increased MAP Kinase signaling and increased proliferation [61,62]. Furthermore, in a study with 48 prostatectomy samples, 15-LOX-1 expression positively correlated with the tumor malignancy [59]. The increased amount of 13-HODE in the GBM cells lines examined suggest that GBM and prostate cancer models may activate a similar pathway for survival.

As transcriptional regulators, PPARs induce a gamut of pathways and feedback loops within cells. In prostate cancer cells, 15-LOX-1 overexpression and 13-HODE lead to PPAR gamma phosphorylation by MAPK signaling, which reduces PPAR gamma activities and increases cell proliferation and tumor growth [60,61,62]. 13-HODE-dependent reduction in PPAR beta/delta induced apoptosis in colorectal carcinoma cells. Meanwhile, the induced overexpression of PPAR beta/delta increased the colonic tumor formation in mice [53,54,63]. The overexpression of 15-LOX-1 and the inhibition of PPAR beta/delta, likely followed by an increase of 13-HODE production and binding to PPAR gamma, were able to reduce colitis-associated colorectal cancer in mice by reducing the phosphorylated signal transducers and activators of transcription 3 (STAT3) [64], which, when highly expressed in GBM, is associated with a significantly reduced patient survival [34]. PPAR beta/delta mRNA expression was detected in all the GBM cell lines examined (Figure 2), yet a lesser degree of expression was observed in GBM cells compared to nontumor brain tissue [34].

PPAR gamma is a known suppressor of the interleukin 6 (IL-6)/STAT3 pathway [65,66], and prostate cancer progression positively correlates with STAT3 phosphorylation [67,68]. According to the qRT-PCR data in Figure 2, PPAR gamma mRNA expression was below detection in U251-MG, U138-MG, and T98G; weakly detected in U87-MG; and very evident in A172. Among GBM samples from TCGA, the PPAR gamma expression was slightly lower (not significant) than the nontumor samples [43]. The use of PPAR gamma agonists in GBM cells is known to reduce brain tumor stemness, cell proliferation, and invasive capacity and, also, by reducing STAT3 activation [69,70,71]. In fact, diabetic patients with GBM using a PPAR gamma agonist showed increased median survival compared with the standard GBM treatments alone [72]. Moreover, there is a very significant increase of STAT3 in GBM cells compared to nontumoral human brain tissue [34]. The current data suggest that the high levels of 13-HODE found in GBM cell lines may be influencing GBM growth in a way that is similar to prostate cancer, by inhibiting PPAR gamma activities and increasing STAT3 activation.

STAT3 is negatively regulated by PTEN (Phosphatase and tensin homolog protein) in GBM [73]. According to Iglesia et al.’s study, in the absence of PTEN, STAT3 performs oncogenic activities with EGFRvIII (Epidermal Growth Factor Receptor Variant III) in the nucleus. Furthermore, a decrease in PTEN expression is negatively correlated with GBM patient survival [41]. According to Gravina et al.’s (2017) study, the U251-MG and A172 cell lines possess mutated PTEN genes, T98G and U138-MG harbor PTEN, and U87-MG is PTEN-deficient [74]. According to the GlioVis database, PTEN and 15-LOX-1 are negatively correlated, while PTEN and 15-LOX-2 are positively correlated [34]. Due to the documented alteration of PTEN expression, a weak expression of 15-LOX-2 (and its products), and apparent expression of 15-LOX-1 (and its products) (Figure 2), investigating the possible relationship between 15-LOX and PTEN should be considered in future studies.

As previously mentioned, treating GBM patients with PPAR gamma inhibitors improved their survival. The inhibitors used in this study, NDGA and luteolin, are commonly used to inhibit the 15-LOX pathway. Luteolin, a dietary flavonoid, is commonly found in broccoli, carrots, and chrysanthemum flowers and is one of the most potent 15-LOX-1 inhibitors [75]. Luteolin not only blocks the H^+^ abstraction from occurring between LA and 15-LOX to form HODEs [76], but it also serves as a natural PPAR gamma agonist [77]. Exogenous 13-HODE augmented cell growth in the U87-MG cells (Figure 3), and the inhibition of 15-LOX through luteolin reduced and even arrested the cell growth (Figure 4 and Figure 5). Thus, the initial influence of 13-HODE on U87-MG cells was restored. This data suggests, as already reported in colon cancer, that 13-HODE might be suppressing PPAR gamma activity in the GBM cells [61,62].

Luteolin is also capable of triggering both intrinsic and extrinsic apoptotic pathways and of stabilizing and accumulating the tumor suppressor protein p53 [78]. In Figure 5, luteolin significantly increased the number of apoptotic events in U87-MG and U251-MG cells. Moreover, luteolin was able to arrest A172 and U87-MG cell cycles in the G2/M phase, an effect correspondingly seen with treatments performed in p53 wildtype colon cancer cells (LoVo cell line) [79]. Despite the subtle differences seen between the p53 mutant and wildtype cell lines, the presence of 13-HODE was found in all GBM cell lines examined. Intriguingly, no significant alteration in cell growth by exogenous 13-HODE or 9-HODE was observed in the p53 mutant T98G.

The viability data obtained in this study with treatments of various oxylipins showed no statistical differences between treatments and the control in A172 and U87-MG, regardless of time and concentration. Although slight decreases in cell viability after 13-HODE (0.5 µM) and 9-HODE (1 µM) treatments were seen in U251-MG, other studies demonstrating the influence of HODEs on cell viability used concentrations greater than 5 µM [41,80]. The highest concentration initially used to examine viability was 1 μM and might not have been enough to create a measurable effect on MTT reactivity. Therefore, the influence of higher concentrations of HODE treatments (1 µM, 5 µM, and 10 µM) was measured with cell counting and trypan exclusion over 72h, and significant increases in viable cells were observed. Both 13-HODE and 9-HODE slightly increased the U87-MG cell number after 72 h, yet 9-HODE’s effect was more pronounced (Figure 3).

### 3.4. Migration/Invasive Capacity

Cell migration was examined to further elucidate the role of 15-LOX metabolites in GBM cells. Since T98G produces 13-HODE and 9-HODE endogenously (Figure 2), and additional exogenous treatments over 72 h did not influence its cell count (Figure 3), cells were then treated with 13-HODE, 9-HODE, or 15-HETE, resulting in a slight decrease (23%) in migration when using 13-HODE (Figure 6). However, when treated with luteolin, T98G and U251-MG cells demonstrated a significant inhibition, and a lesser, but noticeable, inhibition was seen in U87-MG and A172 cells. Wang et al. (2017) recently demonstrated that luteolin inhibits U251-MG and U87-MG cell migration through the p-IGF-1R/PI3K/AKT/mTOR signaling pathway [81]. The data in Figure 6 not only confirm the effect of luteolin on these two cells lines, but they also prove that this occurs in T98G and A172.

The invasion of tumor cells into the surrounding tissue requires the activity of matrix metalloproteases. Metalloproteases are proteolytic enzymes responsible for remodeling the extracellular environment. In many types of invasive tumors, the gelatinases MMP-2 and MMP-9 are often upregulated. Concerning GBM production of MMPs, in vitro, Hagemann et al.’s (2012) review of MMP expression in GBM cell lines showed that MMP-2 was expressed in most cells; however, MMP-9 remained controversially expressed among different studies. The GlioVis database indicated a significant increase in MMP-2, MMP-14, and MMP-9 mRNA in GBM tumor cells compared to normal tissue [41]. Similarly, Wang et al. (2017) identified the presence of MMP-2 and MMP-9 in U251-MG and U87-MG through Western blotting [81]. However, they did not examine MMP-2/MMP-9 activity typically measured by gelatin zymography. Figure 7E,F showed that the metalloprotease MMP-2 was clearly active in U87-MG and T98G, and MMP-9 was absent from the extracellular medium. The HPA found MMP-2 mRNA in the U251-MG, U138-MG, and U87-MG cell lines. Yet, they only found a poor expression of MMP-9 in U138-MG and no expression in U251-MG, U87-MG, or in patient samples (including immunohistochemistry staining) [34]. Since Wang et al. explored the influence and mechanism of luteolin upstream of MMPs [81], they may have measured the intracellular levels of MMPs, which also were reported as having other possible functions [82]. Nevertheless, extracellular MMP-2 and MMP-9 were of interest in this study, and therefore, zymography was chosen to measure the MMP expression/activity.

Concerning the relationship of 15-LOX in GBM and invasive capacity, 13-HODE and 9-HODE treatments induced an increase of MMP-2 mRNA in U87-MG and T98G, respectively (Figure 7D) and did not alter the mRNA expression of the MMP-2-activator MMP-14 (data not shown). Interestingly, the inhibition by luteolin increased the MMP-2 activity in T98G cells, and no significant effect was observed on the MMP-2 activity in U87-MG (Figure 7). As proposed by Wang et al., luteolin’s relationship with MMP expression is through inhibiting an associated pathway. However, the direct relationship of between 15-LOX and MMP-2 remains unknown in cancer. There is one study demonstrating that 15-LOX induces MMP-2 expression in arthritis [83]. Arthritis, like cancer, is sustained through chronic inflammation. One of the upregulated pathways in chronic inflammatory arthritis is the IL-4/STAT6 axis [84]. Conrad and Lu (2000) demonstrated that 15-LOX-1 transcription depends upon the IL-4/STAT6 axis. STAT6, expressed in U87-MG [36], is not only correlated with a reduced patient survival, but it has also been reported as a promotor of U87-MG cell growth and invasion [85]. It is possible that STAT6 in GBM upregulates 15-LOX and its products, thus increasing MMP-2 mRNA.

NDGA, found in the creosote bush plant, is a nonselective redox-active compound and a pan-lipoxygenase inhibitor [86,87]. The inhibition of 15-LOX by NDGA not only reduced cell growth (Figure 4) and induced cell cycle arrest (Figure 5), but it also significantly inhibited cell migration (Figure 6) and completely removed any visible activity of MMP-2 (Figure 7). One caveat of NDGA use as a LOX inhibitor was clearly described by Hernandez-Damian et al. This inhibitor not only hinders LOX activity, but it also is an aggressive reactive oxygen species scavenger and presents other antioxidant properties [88]. Therefore, the effects demonstrated by NDGA could be due to 15-LOX inhibition alone or inhibition coupled with NDGA’s interference in other antioxidant pathways.

## 4. Materials and Methods

### 4.1. Cell Culture

All cell lines were cultured in DMEM (Dulbecco’s modified Eagle’s medium—Gibco, Thermo Fisher Scientific, Walsham, MA, USA) supplemented with 10% (*v*/*v*) FBS (fetal bovine serum—Thermo Fisher Scientific, USA), 50-units/mL penicillin, and 50-μg/mL streptomycin (Thermo Fisher Scientific, USA). All the cells were maintained at 37 °C in a humidified atmosphere with 5% CO_2_ usually in 75-cm^2^ flasks, until the desired confluency. Next, cells were washed with PBS and trypsinized (trypsin 0.025%/EDTA 0.02%, Thermo Fisher Scientific, USA) for further use.

### 4.2. Primer Designer

All primers used in this study were designed using PerlPrimer Software and the sequences deposited at the GenBank (www.ncbi.nlm.nih.gov, Acessed on 04/2014) [89]. The primers used in this study were as follows (forward; reverse):5-LOX: GAAGACCTGATGTTTGGCTACC; AATGTTCCCTTGCTGGACCT12-LOX: CGGAATGAGCAACTTGACTG; TTAGCAGCAGAGACTTTAGGA15-LOX-1: CTGTGAAAGACGACCCAGAG; TCCCGAGCCTGTAAAGACAC15-LOX-2: CTCAATATCAAATACTCCACAGCC; TTTCATCTCATTCAGACTCCTCCFLAP: GAACTGTGTAGATGCGTACCC; GAAGAGTATGATGCGTTTCCCACYSLTR2: CCCTGTCCTCTTCAATCCCT; TTTGCTCCAATCCTTCTCCCBLT1: AGGGACACAAAGAAACATAGAC; ACTTATCACAGGCTTCAAGGABLT2: GGACCCTTCTTTGACTAGAG; CATCACCACCCTCATAATCCLTA4H: GAACACCCATATCTCTTTAGTCAG; CTCCAACAACTAAAGCAATCAGLTC4S: GACGGTACCATGAAGGACGA; AGGAACAGCGGGAAGTACTCCYP4A11: CTTGTCTACCTGTCTCCTACC; GATTCTATCCAAGCCACGAGPPAR alpha: GCACAAATATCCACCACTTTAACC; ATTCGCCGTAATCTTCCCAGPPAR beta/delta: CTGGAGTACGAGAAGTGTGAG; ATTGTAGATGTGCTTGGAGAAGGPPAR Gamma: GACTTCTCCAGCATTTCTACTC; CTTTATCTCCACAGACACGACGPR-132: AAATATGCCAGGGAGGAAGGT; ACGGTGTCAAGAACATGAGG18S: CGGCGACGACCCATTCGAAC; GAATCGAACCCTGATTCCCCGTC

### 4.3. Real-Time qRT-PCR

The extraction of total RNA from cells was carried out with TRIzol reagent (Invitrogen, Thermo Fisher Scientific, MA, USA), followed by ethanol precipitation, as described by the manufacturers. Double-stranded cDNAs were synthesized using 2 µg of total RNA and M-MLV transcriptase (Invitrogen) following the manufacturer’s instructions. Real-time quantitative reverse-transcription PCR was carried out with Syber Green PCR Master Mix (Applied Biosystems, Thermo Fisher Scientific, USA) following the manufacturer’s instructions in a 7300 Real-Time PCR System (Applied Biosystems, Thermo Fisher Scientific, USA). 18S expression was used as a housekeeping gene to normalize RNA expression. The cycle threshold (Ct) of each sample was determined, and the relative expression was calculated using 2^−ΔΔCt^ [90].

### 4.4. LC-ESI-MS/MS

Samples were adjusted to 15% (*v*/*v*) with ice-cold methanol and kept at 4 °C. Forty nanograms of freshly prepared internal standard (12-HETE-d8, Cayman Chemical, Ann Arbor, MI, USA) was added to each sample. After 15 min, the sample was centrifuged (3000 rpm for 5 min), the precipitated proteins were removed, and the supernatant was acidified to pH3.0 with 0.1-M hydrochloric acid and immediately placed on C18 SPE cartridges preconditioned with 20 mL of methanol and 20 mL of water. The columns were washed with 20 mL of 15% methanol (Merck, São Paulo, Brazil), 20 mL water, and 10 mL of hexane. Finally, the bioactive lipids were eluted in 15 mL of methyl formate. The solvent was evaporated under nitrogen in the dark, and the residue was dissolved in 100 µL of 70% (*v*/*v*) ethanol (Merck, São Paulo, Brazil) to be injected into the LC-MS/MS.

The Thermo Accela TSQ Quantum Max LC-MS/MS apparatus was operated in the electrospray negative ionization mode. Calibration lines were run for each of the 21 oxylipins, containing from 1 pg/µL to 200 pg/µL. Optimal conditions for each individual (S)-form lipid were determined by direct infusion of a standard solution of 10ng/µL. Typical parameters for use were: spray voltage 3500V, discharge current 80V, capillary temperature 233 °C, vaporizer temperature 360 °C, collision energy 13–25 V, tube lens 91–119 V, and scan time 0.2 s.

A C18 Kinetex 2.1 × 100 mm, 2.6 µM chromatography column (Phenomenex, Torrens, CA, USA) was used, and the sample volume was 5µL at a flow rate of 200 µL/min with samples kept at 8 °C. Samples were run using Solvent A—acetonitrile/H_2_O/acetic acid (45/55/0.02—*v*/*v*/*v*) and solvent B—methanol/H_2_O/acetonitrile (80/20/0.02—*v*/*v*/*v*), with a gradient as follows: 0–1 min, 50% A:50% B; 1.01–20 min, 50%–10% A:50%–90% B; 20.01–25 min, 10%–0% A:90%–100% B; 25.01–28 min, 0% A:100% B; 28.01–28.10 min, 50% A:50% B; and 28.10-end 50% A:50% B. Results were analyzed using Thermo XCalibur software.

### 4.5. MTT (3-(4,5-Dimethylthiazol-2-yl)-2,5-Diphenyltetrazolium Bromide) Assay

Cells were seeded at 3 × 10^3^ for U251-MG and U87-MG and at 5 × 10^3^ for A172 in 96-well plates. After 24 h, cells were treated with 0.1 µM, 0.5 µM, and 1 µM of the following (S)-form oxylipins: 13-HODE, 9-HODE, 15-HETE, 5-HETE, 12-HETE, 8-HETE, 9-HETE, 20-HETE (all Cayman Chemical, USA) and 100% ethanol (control). At 24, 48, and 72 h of treatment, each well was incubated for 4 h with 0.25 mg/mL of tetrazolium at 37 °C in a humidified atmosphere with 5% CO_2_. At the end of incubation, cells were washed with warm PBS and lysed with 100 µL of 0.04-M HCl in isopropanol to solubilize the formazan. The absorbance was read at 590 nm in an Epoch microplate reader (BIOTEK, Weinusky, VT, USA).

### 4.6. Treatmentd with 13-HODE, 9-HODE, and 15-HETE

To determine their impact on GBM growth in vitro, exogenous (S)-forms 13-HODE, 9-HODE, 15-HETE, and their controls (100% ethanol) were added to the cell lines in varying concentrations. Twelve hours after seeding 3 × 10^4^ for U87-MG and T98G cells in 24-well plates, the culture medium was changed to a medium containing different concentrations of 9-HODE, 13-HODE, or 15-HETE. The cell medium was changed every 24 h with fresh treatments. To measure the influence of the treatments on culture growth and viability, the cells and medium were collected at 24 h, 48 h, and 72 h and stained with 0.4% trypan blue (Merck, São Paulo, Brazil) to count both viable and unviable cells in a Neubauer chamber. All treatments were tested at least three times in triplicates.

### 4.7. Lipoxygenase Inhibitor Treatments

Cells were seeded at 2 × 10^4^ for U251-MG and U87-MG and at 3 × 10^4^ for A172 in a 24-well plate. After 24 h, the cells were treated with pharmacological inhibitors luteolin (15-LOX), nordihydroguaiaretic acid—NDGA (12-LOX and 15-LOX), CAY10606 (5-LOX), CAY10649 (5-LOX), or MK886 (FLAP binding) (all Cayman Chemical, USA). The vehicle, DMSO, was used as a control. Inhibitor-containing cell medium was replaced every 24 h until 72 h of treatment. At the end of 72 h, each well was washed with PBS, trypsinized, and the total number of viable, as defined by trypan blue exclusion, was counted using a Neubauer chamber.

### 4.8. Cell Cycle Assay—Propidium Iodide Fluorescence

Cells were seeded at 2 × 10^4^ for U251-MG and U87-MG and at 3 × 10^4^ for A172 in a 24-well plate. After 24 h, the cells were treated with luteolin or NDGA inhibitors, as previously described. After 72 h of treatment, cells in each well were trypsinized, followed by centrifugation at 1500 rpm for 5 min. The cell pellet was rinsed with ice-cold PBS, centrifuged, and resuspended in ice-cold 70% ethanol for cell fixation for at least 24 h. After fixation, cells were again washed with ice-cold PBS and incubated for 30 min with 500 μL of a staining solution (20-μg/μL propidium iodide, 50-μg/μL RNase A, Thermo Fisher Scientific, USA, and 0.1% Triton X-100, Merck, São Paulo, Brazil). At the end of incubation, cells were centrifuged, resuspended in ice-cold PBS, and kept on ice for reading. Cell cycle phase was measured through propidium iodide detection in a Guava^®^ easyCyte flow cytometer (Millipore, Burlington, MA, USA) using 532 nm as the excitation wavelength (λ_ext._).

### 4.9. Cell Death Assay—Annexin V—Propidium Iodide Fluorescence

Cells were seeded at 12 × 10^4^ for U251-MG and U87-MG and at 3 × 10^4^ for A172 in 12-well plates. After 24 h, the cells were treated with luteolin or NDGA inhibitors, as previously described. After 72 h of treatment, cells in each well were trypsinized, followed by centrifugation at 1500 rpm for 5 min. Next, 1 × 10^5^ cells were resuspended in 100 µL of 1X reaction buffer (5X reaction buffer: 50-mM HEPES, 700-mM NaCl, and 12.5-mM CaCl_2_, pH 7.4) containing 5 µL of Alexa Fluor^®^ 488 annexin V (Thermo Fisher Scientific, Inc., USA) and 1 µL of propidium iodide solution (100 µg/mL of propidium iodide in 1X reaction buffer). Cells were incubated for 15 min at room temperature. For standardization, samples of cells were also incubated with annexin V or propidium iodide alone. At the end of incubation, 400 µL of 1X reaction buffer were added to each sample before green and red fluorescence detection in a Guava^®^ easyCyte flow cytometer (Millipore, USA) using a λ_ext_. of 488 nm for Alexa Fluor^®^ 488 annexin V detection and λ_ext._ of 532 nm for propidium iodide detection.

### 4.10. Wound Healing Assay

T98G cells were seeded at 9 × 10^4^ cells in a 24-well plate to reach 90–100% confluency in 12 h. After this time, using a 200-µL pipette tip, a streak was made in the middle of the well. The medium was carefully changed to include previously determined concentrations of OXLAMs and drugs with their respective controls. Pictures were taken immediately (0 h) and after 12 h. First, the difference of the cells invading the wound between times 0 and 12 h of each treatment were calculated. Then, the differences between the treatments and their respective controls were calculated using ImageJ Program tools and GraphPad Prism 5 Software.

### 4.11. Transwell Migration Assay

The Transwell migration assay was performed using a 24-well plate with 8.0-μm pore polycarbonate membrane Transwell inserts (Corning, NY, USA). Initially, U251-MG, U87-MG, and A172 cells were seeded onto the top compartment at 2 × 10^4^ for U251-MG and U87-MG and at 3 × 10^4^ for A172. Cells were kept at 37 °C in a humidified atmosphere with 5% CO_2_ for 6 h until treatment with luteolin or NDGA inhibitors. After 24 h of treatment, the inserts were washed with PBS; then, cells in the upper compartment were physically detached with a cotton swab. The inserts with remaining cells in the bottom compartment were stained with crystal violet in 10% ethanol (Merck, São Paulo, Brazil). Polycarbonate membranes were mounted on glass slides with Entellan (Merck/Millipore), and images of 10 different random fields per membrane were captured with a coolSNAP-Pro camera (Media Cybernetics, Rockville, MD, USA) attached to a Nikon Optiphot-II microscope.

### 4.12. Western Blot

The 10% gels used for electrophoresis were prepared and placed inside the electrophoresis cube with fresh running buffer. Forty micrograms of protein from each sample were added to the wells. Fresh running buffer was prepared and used (Glycine + UltraPure^TM^ Tris (Invitrogen, Thermo Fisher Scientific, USA) + H_2_O, pH 8.3). After electrophoresis, the gel was immediately placed in the transfer apparatus, and the samples were transferred onto a nitrocellulose membrane in fresh transfer (blotting) buffer (TBS 1X + Methanol Merck, São Paulo, Brazil). The transfer was confirmed by staining the membrane with Ponceau red solution (Merck, São Paulo, Brazil).

After washing the membranes with TBS 1x to remove the stain, 5% fat-free Milk + TBS Tween 20 (Merck, São Paulo, Brazil) was used to block for 1 h. After removing excess milk with a wash in TBS 1x, the primary antibodies (15-LOX-1 (1:700) R & D Systems, USA, MMP-2 (1:2000) R&D Systems, MMP-14 (1:2000) R&D Systems, and actin (1:3000) Abcam, USA) were added to the membranes and incubated overnight in 4 °C. The next morning, the membranes were washed, and the horseradish peroxidase (HRP)-labeled secondary antibodies were added (anti-sheep (15-LOX-1), anti-goat (MMP-2), anti-mouse (MMP-14), and anti-rabbit (actin) (1:2000)) and incubated for 2 h at room temperature. In the case of 15-LOX-1, the secondary antibody possessed a biotinylated conjugate requiring an additional 3.5-h incubation with streptavidin HRP conjugate (ExtrAvidin peroxidase; Merck, São Paulo, Brazil) in order to have a reaction with the ECL solution. Finally, the membranes were washed and then revealed with enhanced chemiluminescence (ECL, Bio-Rad Laboratories Inc., São Paulo, SP, Brazil) in a Syngene G-BOX with its GeneSys program. The ImageJ program was used to determine relative band intensities. 

### 4.13. Zymography

The 10% gelatin (Synth, São Paulo, Brazil) zymogram was prepared and stored in running buffer, as recommended by Toth and Fridman [91]. Fresh serum-free samples were removed from cell culture after 21 h of incubation and stored on ice. Samples were kept in a −80 °C freezer (no more than 1 week) before they were slowly thawed on ice for 2 h for further use. The gels were placed in a running chamber with running buffer while the samples thawed. Once thawed, the samples were centrifuged at 10,000× *g* for 5 min at 4 °C. The supernatant was separated and used for the zymogram. Samples were diluted 1:1 with 2× sample (Laemmli) buffer. After 5 min, 25 µL of the sample were placed in the well of the loading gel. The gel ran for 2 to 3 h at 100 V. The gels were washed with Triton X-100 (Merck, São Paulo, Brazil) buffer three times for 15 min on an agitator. Then, the gels were incubated at 37 °C for 17 h in development buffer (Tris-HCl, pH 8.8, CaCl_2_, NaN_3_, and ddH_2_O). The negative control was a second gel washed with Triton X-100 containing a chelating agent (EDTA (10 mM)) and was incubated with development buffer containing 10-mM EDTA. The next day, the gels were washed with fixation buffer (ddH_2_O, methanol, and acetic acid 45/45/10 (*v*/*v*/*v*)) and stained using 0.1% Coomassie blue staining buffer (Merck, São Paulo, Brazil) for 30 min. Then, the gels were destained using the fixation buffer until bands could be seen. Images were captured in a Syngene G:BOX—GeneSys program (Daly City, CA, USA).

### 4.14. Statistical Analysis

All data were plotted and analysed using GraphPad Prism 5.0 (GraphPad Prism 5 Software, San Diego, CA, USA). Values express the arithmetic mean ± standard error. Analysis between two groups was performed with Student’s *t*-test. Analysis between three or more groups, considering only one variable, was performed with one-way ANOVA followed by Dunnett’s test. Analysis between three or more groups, considering two variables, were performed with two-way ANOVA followed by Bonferroni test. The differences were considered statistically significant at *p* < 0.05.

## 5. Conclusions

Similar to the roles identified in prostate cancer, our data together corroborate our hypothesis that 15-LOX and its OXLAM products, 13-HODE and 9-HODE, generate an overall protumoral effect on GBM cell growth and migration. The inhibition of 15-LOX and the use of a natural flavonoid, luteolin, present new avenues for targeting the 15-LOX pathway to compliment GBM therapies.

## Figures and Tables

**Figure 1 ijms-21-08395-f001:**
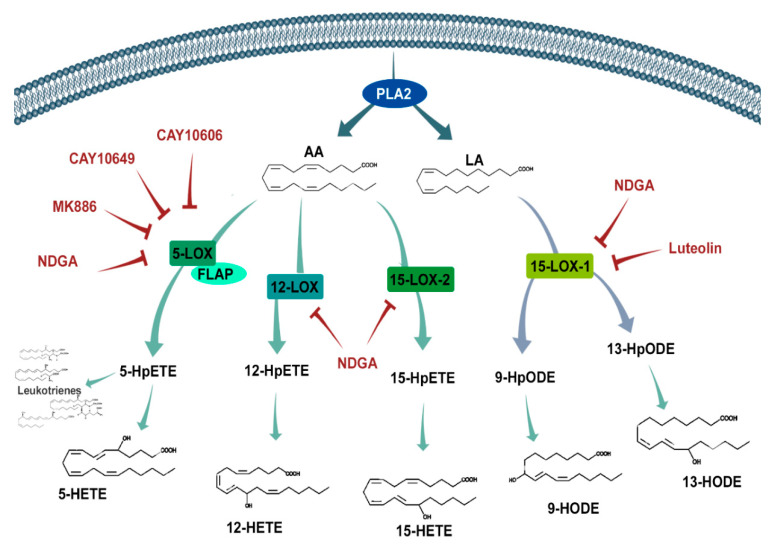
Schematic representation of the lipoxygenase pathway and its inhibitors. Phospholipase A2 (PLA2) releases arachidonic acid (AA) and linoleic acid (LA) into the cytosol from the cell membrane, where they are exposed to oxygenation by different lipoxygenases (5-LOX, 12-LOX, and 15-LOX) to produce intermediate hydroperoxyeicosatetraenoic acid (HpETE) and hydroperoxyoctadecadienoic acid (HpODE) products. After further oxidation, these bioactive products (hydroxyeicosatetraenoic acids (HETEs), hydroxyoctadecadienoic acids (HODES), and leukotrienes) interact with several receptors influencing different signaling cascades that are known to be involved in cell cycle, survival, and migration. The inhibitors (red text) used in this study aimed to prohibit these LOXs from synthesizing their products. FLAP—Five LOX-Activating Protein and NDGA—nordihydroguaiaretic acid.

**Figure 2 ijms-21-08395-f002:**
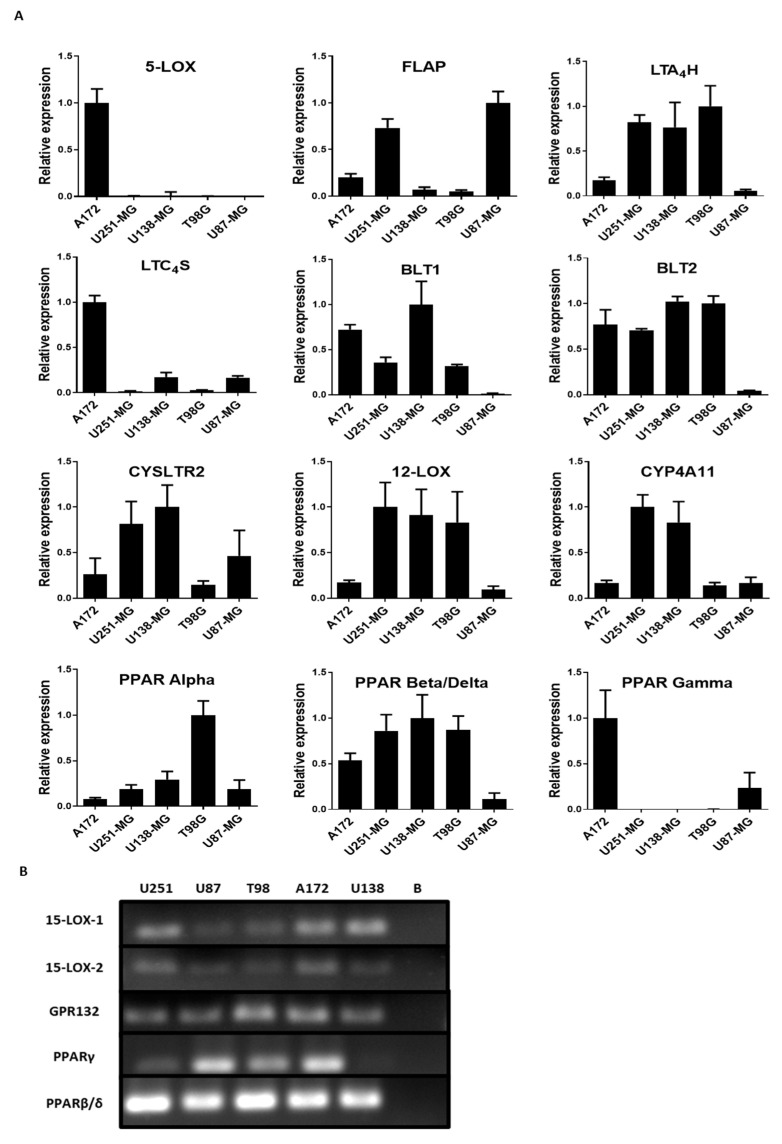
Profile of the lipoxygenase pathway metabolism in 5 glioblastoma (GBM) cell lines. (**A**) qRT-PCR of the enzymes and receptors involved in lipoxygenase and cytochrome P450 (CYP450) pathways in a GBM cell line. Histograms show the relative expression of the genes for: 5-LOX, FLAP, LTA_4_ hydrolase (LTA4H), LTC_4_ synthase (LTC4S), BLT1, BLT2, CYSLTR2, 12-LOX, CYP4A11, peroxisome-proliferator-activated receptor (PPAR) alpha, PPAR beta/delta, and PPAR gamma. Values represent the genetic expression relative to the endogenous 18S and the standard deviations calculated by ΔΔCT. N = 3, in duplicate. (**B**) PAGE of conventional RT-PCR of the 15-LOX pathway in five GBM cell lines. Cell lines U251-MG, U87-MG, T98G, A172, and U138-MG were used to identify the genes associated with the 15-LOX pathway. All the genes were amplified in all cell lines, except PPARγ (gamma) in U138-MG. Blank (**B**).

**Figure 3 ijms-21-08395-f003:**
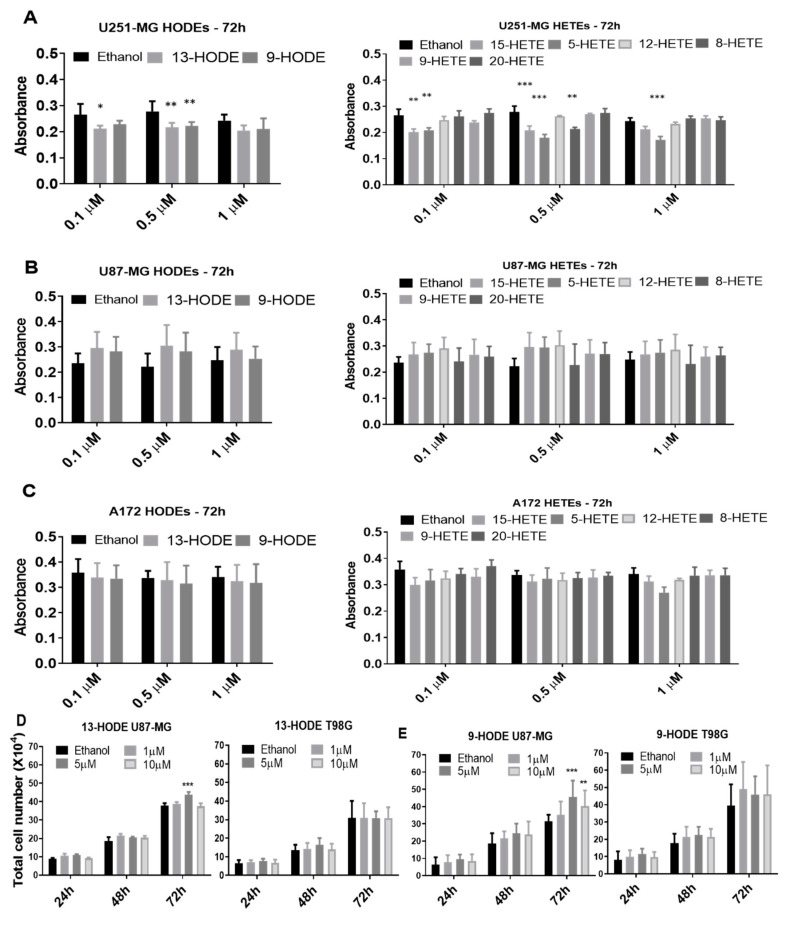
Cell viability of cells treated exogenously with lipids as measured by 3-(4,5-dimethylthiazol-2-yl)-2,5-diphenyltetrazolium bromide (MTT) conversion (**A**–**C**). Histograms show MTT absorbance after 72 h of treatment with a HODE and HETE panel for U251-MG (**A**), U87-MG (**B**), and A172 (**C**). Significance was found only in some treatments of U251-MG (A). Exogenous 13-HODE and 9-HODE treatments on U87-MG and T98G (**D,E**). U87MG treated with 13-HODE demonstrated a significant increase in cell count with the 5 µM treatment at 72 h, while no effect was seen on T98G (**D**). U87MG treated with 9-HODE demonstrated a significant increase in cell count with 5 µM and 10 µM treatments at 72 h, while no effect was seen on T98G (**E**). *p* < 0.05 (*), *p* < 0.01 (**), and *p* < 0.001 (***). N = 3, in duplicate.

**Figure 4 ijms-21-08395-f004:**
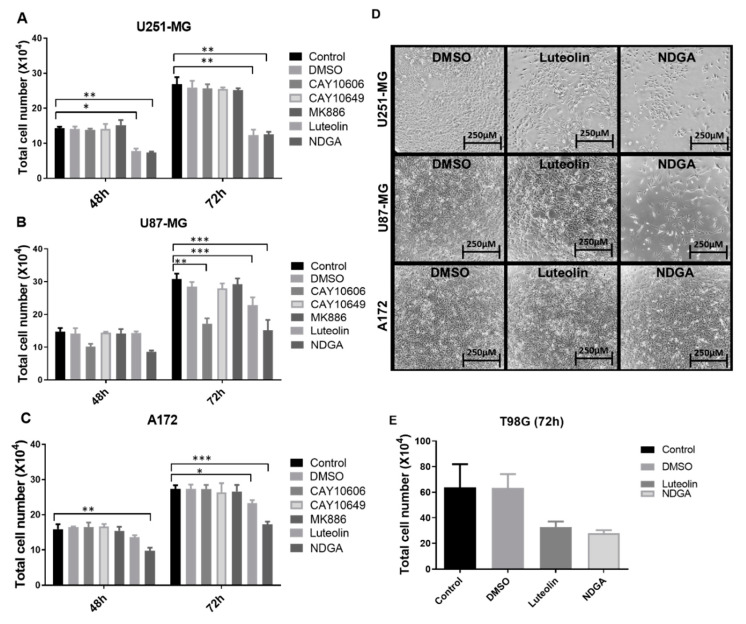
Treatment with the inhibitors luteolin, NDGA, CAY10606, CAY10649, MK886, CAY10678, and CAY10526 in U87-MG (**A**), U251-MG (**B**), and A172 (**C**). Graphs show the results after 48 h and 72 h, with the data analysis performed in relation to the control; cell numbers decrease after NDGA and luteolin. Phase-contrast microscope images show cell plates at 72 h right before collection (**D**). Luteolin and NDGA treatment at 72 h with T98G (**E**). * *p* < 0.05, ** *p* < 0.01, and *** *p* < 0.001. N = 3, in duplicate, except for (**E**), which is N = 2 in duplicate. DMSO: dimethyl sulfoxide.

**Figure 5 ijms-21-08395-f005:**
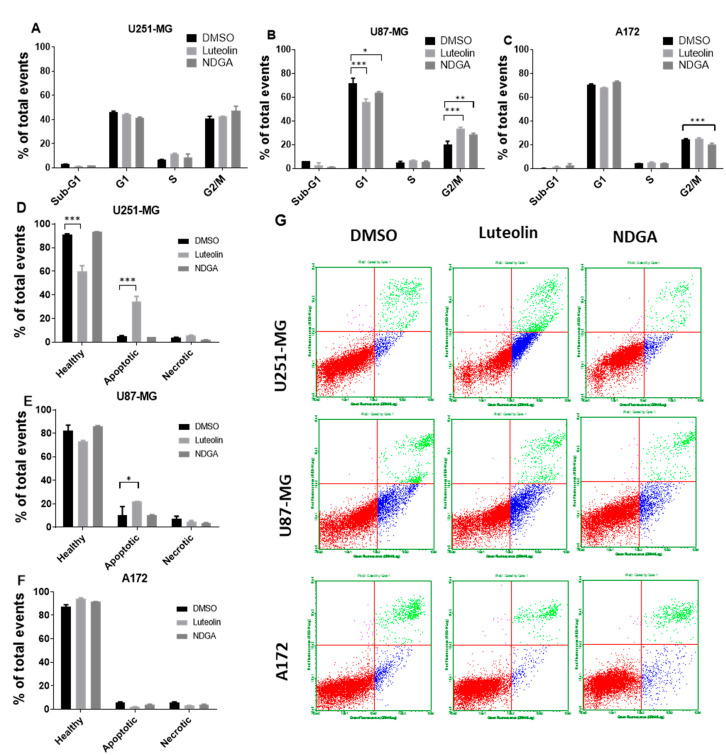
Flow cytometer analysis of the cell cycle and apoptosis. Cells were treated with luteolin and NDGA for 72 h before propidium iodate incorporation. Histograms show the percentage of U251-MG (**A**), U87-MG (**B**), and A172 (**C**) cells in each cell cycle phase. Both luteolin and NDGA reduced G1 and increased G2/M in the U87-MG population. Luteolin did not influence the U251-MG or A172 cell cycles. Apoptosis analysis by flow cytometer. (**C**) cells were treated with luteolin and NDGA for 72 h before propidium iodate and annexin V incorporation. Histograms show the percentage of apoptotic cells in U251-MG (**D**), U87-MG (**E**), and A172 (**F**), with an increase in apoptotic cells both in U251-MG and U87-MG. Dot plot graphs (**G**) show population distribution among healthy (red), apoptotic (blue), and necrotic (green) cells. * *p* < 0.05, ** *p* < 0.01, and *** *p* < 0.001. N = 2, in duplicate.

**Figure 6 ijms-21-08395-f006:**
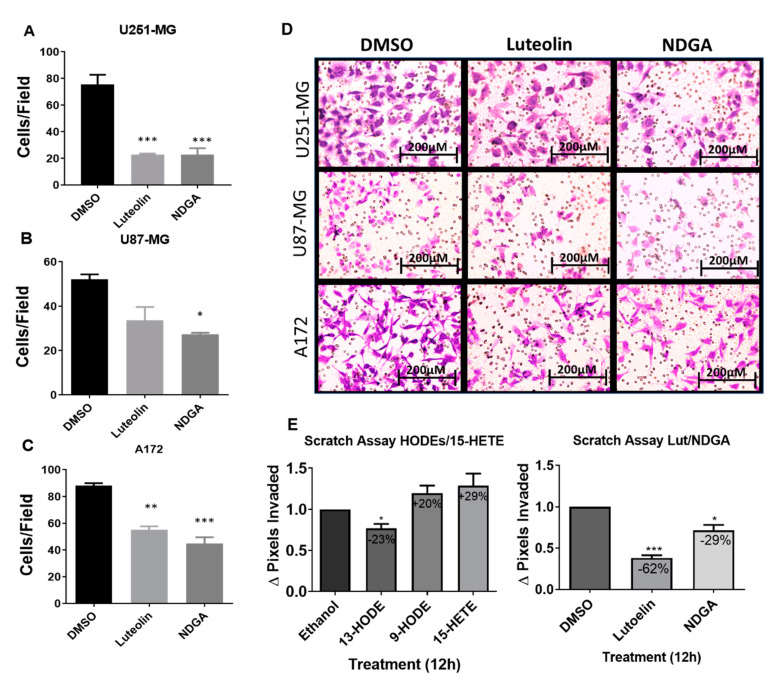
Transwell migration of the GBM cell line after treatment with the pharmacological inhibitors. Histograms expressing the cell number after 8-μm pore migration show a decrease in Transwell migration after luteolin and NDGA treatments (**A**–**C**). The values represent the arithmetic mean of numbers that were individually obtained with the mean of ten different visual fields ± standard error. Representative images of the cells after 24 h of migration stained with violet crystal (**D**). 100× magnification, captured by light microscopy. N = 2, in duplicate. Wound-healing assay with cell line T98G. Histograms show the effects of 5 µM treatments of 13-HODE, 9-HODE, and 15-HETE (left) and 15 µM treatments of 15-LOX inhibitor (luteolin) and 40 µM treatments of 15/12-LOX inhibitor (NDGA) (right) on T98G cell migration (**E**). A decrease in migration was observed in the 13-HODE treatment compared to its vehicular control (ethanol). An inhibitory effect was observed with the luteolin and NDGA treatments compared to their control, dimethyl sulfoxide (DMSO). N = 3. * *p* < 0.05, ** *p* < 0.01, and *** *p* < 0.001.

**Figure 7 ijms-21-08395-f007:**
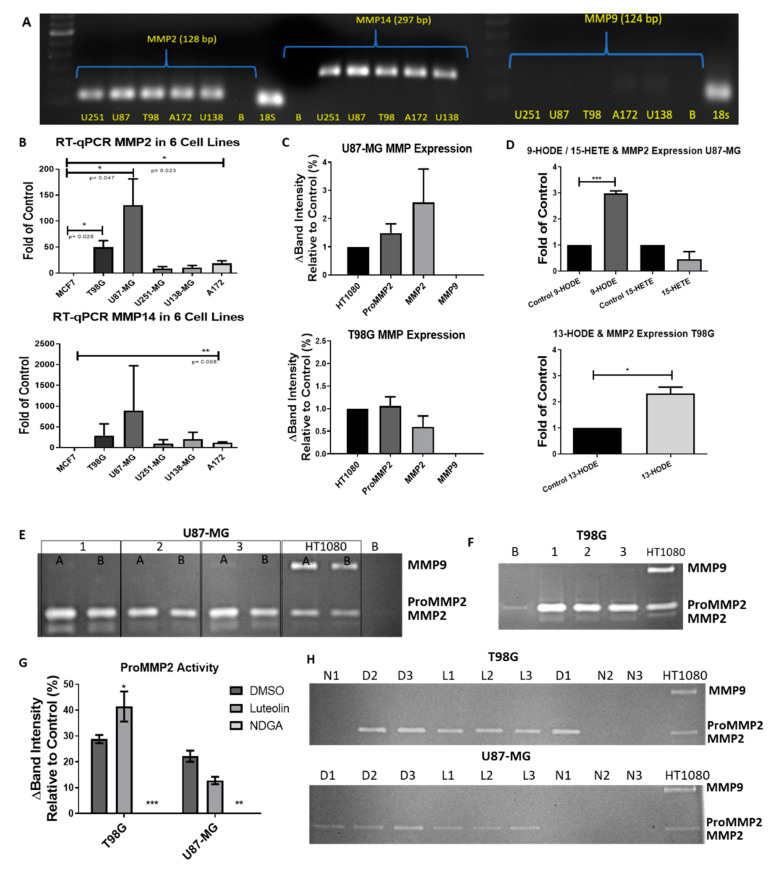
RT-PCR/zymography basal metalloprotease (MMP) expression in GBM cell lines. (**A**) RT-PCR gel electrophoresis of MMP-2, -9, and -14 in 5 GBM cell lines. MMP9 was poorly expressed. (**B**) qRT-PCR of the relative MMP expression in 5 GBM cell lines compared to MCF7. (**C**) Zymographies of baseline MMP expression in U87-MG (top) and T98G (bottom) cells were obtained by cultivation in T-25 plates with 2 mL of serum-free Dulbecco’s modified Eagle’s medium (DMEM) for 21 h. Relative intensities were adjusted by the number of cells counted in each plate when the medium was collected and compared with the HT1080 relative expression. (**D**) qRT-PCR expression of MMP2 mRNA after 72 h of treatments with HODEs and 15-HETE in U87-MG (top) and T98G (bottom). (**E**) Gelatin zymography of U87-MG samples cultivated in either 2 mL (Lane 1A) or 4 mL (Lane 1B) of the medium were run in duplicates. Each lane was a separately cultivated trial. Each band shows the relative activity of proMMP2 (72 kD), activated MMP2 (MMP2: 62 kD), and proMMP9 (92–82 kD) compared with the HT1080 fibrosarcoma cell line. (**F**) Zymography of the T98G cell medium. (**G**) Zymography values of proMMP2 expression after 72 h of treatment with 15-LOX inhibitors. (**H**) Zymography gels of T98G (top) and U87-MG (bottom). *D1*, *D2* and *D3* = (*DMSO N* = *1*, *2* and *3*). *L1*, *L2* and *L3* (luteolin *N* = *1*, *2* and *3*). *N1*, *N2* and *N3* (NDGA *N* = *1*, *2* and *3*). B = blank. All experiments with N = 3.

**Figure 8 ijms-21-08395-f008:**
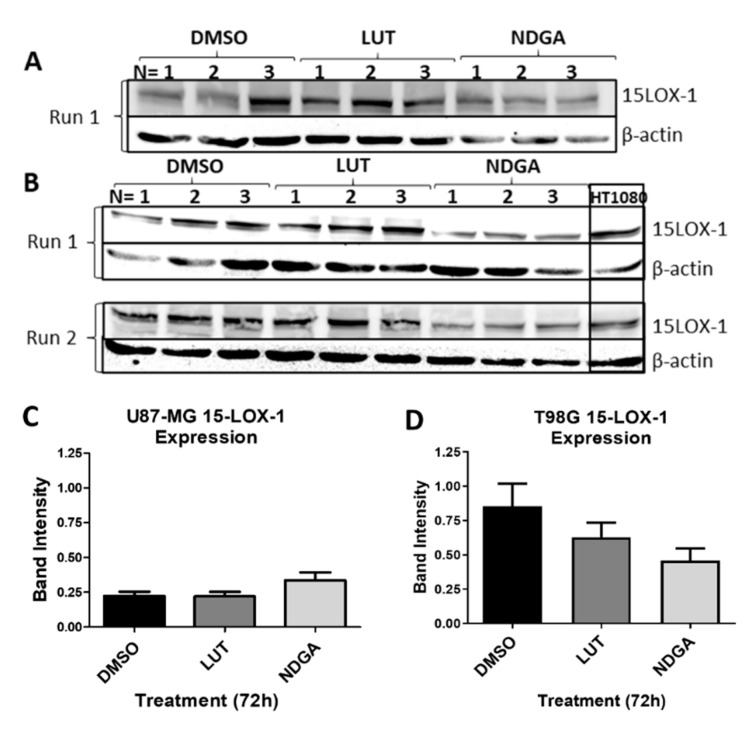
Western blots of 15-LOX-1 expression after treatments with LOX inhibitors luteolin (LUT) and nordihydroguaiaretic acid (NDGA) for 72 h. (**A**) U87-MG protein samples of treated cells. (**B**) T98G protein samples of treated cells. The Western blot protocol was performed 2 times using fresh samples and new nitrocellulose membranes (Run 1 and Run 2). Each membrane possesses three separate trials of cells incubated in each treatment (*N* = *1*, *2* and *3*). The HT1080 cell line was used as a positive control. (**C**) U87-MG and (**D**) T98G statistical analyses. 15-LOX-1 values were normalized to the expression of each sample’s respective endogenous control, β-actin. An unpaired *t*-test with Welch’s correction was performed, and a *p*-value ≤ 0.05 was considered significant.

**Table 1 ijms-21-08395-t001:** Oxylipins produced by the lipoxygenase pathway in cultured glioblastoma (GBM) cell lines in nanomolar concentrations. Liquid chromatography-electrospray ionization-tandem mass spectrometry (LC-ESI-MS/MS) analysis of the products 13-HODE (hydroxyoctadecadienoic acid), 9-HODE, 15-HETE (hydroxyeicosatetraenoic acid), 17-HDHA (hydroxy docosahexaenoic acid), 12-HETE, and 5-HETE. N = 3.

	A172	U87-MG	U138-MG	U251-MG	T98G
**9-HODE**	0	0	0.608 + 0.013	2.090 + 1.89	0.270 + 0.02
**13-HODE**	1.094 + 0.52	2.253 + 1.01	3.658 + 2.57	1.610 + 0.85	1.711 + 1.03
**5-HETE**	0.317 + 0.13	0	0	0	0
**12-HETE**	0	1.712 + 0.34	0	2.928 + 0.11	0
**15-HETE**	0	0	0	0	1.062 + 0.55
**17-HDHA**	3.86 + 1.98	0	0	0	0.940 + 0.43

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
