# Peer review of "Influence of Lipoxygenase Inhibition on Glioblastoma Cell Biology"

_ijms, 2020, doi:10.3390/ijms21218395_

Round 1

Reviewer 1 Report

I have read the manuscript with interest. The idea and the perspectives are novel and the presentation scientifically sound.

I only have few comments:

  • the title does not reflect the content of the article
  • I'd like to see a diagram showing interactions between the studied genes and their inhibitors 
  • In Fig. 2 A, B, C, D on the y-axis should be the same scale (maximum value 0.4 or 0.5)
  • cell line names should be uniform (now U87 and U87-MG are interchangeable; the same applies to the other cells)
  • In fig 5D scale is missing

Author Response

Responses to Reviewer 1 – Manuscript: ijms-979494

Dear Reviewer #1,

Thank you for taking the time to carefully review our work. We appreciate your comments and suggestions. Please refer to the responses to your concerns written below:

Point 1: The title does not reflect the content of the article

RESPONSE: Thank you for your suggestion, we have changed the title to Influence of lipoxygenase inhibition on glioblastoma cell biology.

Point 2: I'd like to see a diagram showing interactions between the studied genes and their inhibitors

RESPONSE: Thank you for this idea. We have added a schematic representation as the new Figure 1 to help visualize the pathway.

Point 3: In Fig. 2 A, B, C, D on the y-axis should be the same scale (maximum value 0.4 or 0.5)

RESPONSE: Thank you for showing us this item. We have corrected the scales to have maximum values of 0.5.

Point 4: Cell line names should be uniform (now U87 and U87-MG are interchangeable; the same applies to the other cells)

RESPONSE: Thank you for identifying this inconsistency. We have corrected the text, and in the figures where the more abbreviated versions are observed (due to space constraints) we have clarified their legends.

Point 5: In fig 5D scale is missing

RESPONSE: Thank you for identifying this missing information. We have added scales to the figures containing micrographs.

Reviewer 2 Report

This manuscript reported a study of the effects of the metabolites and inhibitors of 15-LOX on the growth, invasion of glioblastoma cell lines. The design of the experiments was poorly described and the interpretation of the experimental data was not clear. There are several major issues in the current form of the manuscript. The data generated also showed no clear theme of the problems that the authors were planning to solve. As presented in the manuscript, it reads like a compilation of several loosely related experimental data without any clear message that is related to the topic. The points were discussed in the following.

  1. The choices of the cell lines were confusing. Line 135 mentioned that U251-MG, U87-MG and 172 were selected for study. However, the T98-G cell line was used for comparison in Figs. 2 and 3. What is the reason of using T98-G as the control?
  2. Based on the profiles listed In Table 1, the 5 cell lines are heterogenous and their dependence to the lipoxygenase pathway varies. The authors did not describe any hypothesis related to the upstream and downstream effectors in the lipoxygenase pathway that may be important to the survival of these GBM cell lines. If the lipoxygenase pathway is important for this study, the authors should use the information in Table 1 to explain why certain cell lines will be dependent on specific metabolites and show the data to support it.
  3. Why the cell lines are selectively responsive to different 15-LOX inhibitors in Figure 3? How selective are these inhibitors used in the study?
  4. What are the targeting proteins of 9-HODE, 13-HODE, 15-HOTE? Do they overlap with the targets of Luteolin and NDGA in the data shown in Fig. 5?
  5. How and why the perturbation in the Lipoxygenase pathway affects the MMP-2 and 9 expressions in Fig. 6? What is the role of MMP-14 in Fig.6? Why breast cancer cell line MCF-7 and sarcoma cell line HT1080 were included in Fig. 6? What is control-3H and 6-H in Fig. 6?

Author Response

Responses to Reviewer 2 – Manuscript: ijms-979494

Dear Reviewer #2,
Thank you very much for the insight, questions, and suggestions. Our responses to your concerns are as follows:

Point 1: The choices of the cell lines were confusing. Line 135 mentioned that U251-MG, U87-MG and 172 were selected for study. However, the T98-G cell line was used for comparison in Figs. 2 and 3. What is the reason of using T98-G as the control?

RESPONSE: Thank you for addressing this issue. We have adjusted the sentence starting on Line 146 (previously 135) to state “Thus, based on the mRNA and LC-MS/MS profile, three cell lines were initally chosen to represent GBM cells with a 15LOX/13-HODE profile: U251-MG, U87-MG and A172. However, as the study progressed, we identified T98G as another cell line of interest due to its detectable production of both 13-HODE and 15-HETE as well as its history of being a more drug resistant cell line.” Therefore, we began to insert T89G in the other assays as an additional cell line and not as a control. We have also reiterated this concept in a sentence beginning on Line 194.

Point 2: Based on the profiles listed In Table 1, the 5 cell lines are heterogenous and their dependence to the lipoxygenase pathway varies. The authors did not describe any hypothesis related to the upstream and downstream effectors in the lipoxygenase pathway that may be important to the survival of these GBM cell lines. If the lipoxygenase pathway is important for this study, the authors should use the information in Table 1 to explain why certain cell lines will be dependent on specific metabolites and show the data to support it.

RESPONSE: Thank you, we found your question interesting. To the extent of our knowledge, this study is the first of its kind exploring the role of the LOX pathway in GBM. The five human GBM cell lines were chosen to represent the heterogenous nature of GBMs. Moreover, the cell lines presented in Table 1 clearly demonstrate the heterogeneity of these cells in their endogenous production of lipoxygenase products. However, due to the exploratory nature of this study, the relationship between the LOX pathway and GBM can only be defined by the influence of the pathway on certain cellular activities. Further studies involving genetic interventions (knock-down, knock-out) or in vivo applications are required to adequately define the importance of this pathway. Our study will open the way for more investigations.

Point 3: Why the cell lines are selectively responsive to different 15-LOX inhibitors in Figure 3? How selective are these inhibitors used in the study?

RESPONSE: Thank you for your question. As referred to in our response to question 2, we are attempting to represent the heterogenous nature of human GBM tumors through the different cell lines selected. As seen in table 1, the baseline production of the products of different LOXs in the cell lines vary; therefore, we are not surprised by the varied responses to the inhibitors. Concerning the specificity of Luteolin please refer to Lines 448-451 and references 75-77, and concerning NDGA, please refer to Lines 518-525 for a brief description and references 86-88.

Point 4: What are the targeting proteins of 9-HODE, 13-HODE, 15-HOTE? Do they overlap with the targets of Luteolin and NDGA in the data shown in Fig. 5?

RESPONSE: Thank you for your question. We believe that Lines 70-73, in the introduction, give a clear answer to your question concerning target proteins (which are their respective receptors). Since the role of the inhibitors is to block the LOX’s synthesis of HODEs/HETEs from LA and AA, we do not believe that there is any direct interaction between the inhibitors and the products of the enzymes they inhibit. We have added a schematic representation (Figure 1) of the pathway to help readers.

Point 5: How and why the perturbation in the Lipoxygenase pathway affects the MMP-2 and 9 expressions in Fig. 6? What is the role of MMP-14 in Fig.6? Why breast cancer cell line MCF-7 and sarcoma cell line HT1080 were included in Fig. 6? What is control-3H and 6-H in Fig. 6?

RESPONSE: These are good questions. We do not know how the LOXs are influencing MMPs nor why they are influencing MMP expression (Line 499 begins a brief discussion of this). However, we have proven that LOX does influence MMP expression in GBM. The MCF-7 and HT1080 cell lines serve as biological/experimental controls since they are known to produce the MMPs investigated in this study. Control-13H represents Control of 13-HODE and this applies to Control-9H (9-HODE) and Control-15H (15-HETE). This was clarified in the figure (now Figure 7) as Control 13-HODE, Control 9-HODE, Control 15-HETE, respectively, thank you.

Round 2

Reviewer 2 Report

The authors have addressed most concerns and made correction. Fig. 1 in the revised manuscript provided a clearer ideas for understanding the action of the inhibitors used in the manuscript. 

Regarding the response to point 5, it is difficult to compare different cancer cell lines treated with the same inhibitor. Breast cancer cells may have different genetic alterations not shared by glioblastoma cell lines such as the regulation of MMP2 and MMP9 expression. The authors should make a cautionary note. However, a recent paper linking MMP2/9 with 15-lipoxygenase/15-HETE mediated via hypoxia activation may be of interest to the authors.

https://www.sciencedirect.com/science/article/pii/S0022282818303328?via%3Dihub

The study may suggest the different effects on the MMP2/9 levels in the treatment between Luteolin and NDGA in Fig. 7.

There are likely other references in different contexts and the authors are suggested to do literature search.

Author Response

Response to Reviewer 2:
Thank you again for your contribution to improving this manuscript. Here are our responses to your concerns.

Point 1: Regarding the response to point 5, it is difficult to compare different cancer cell lines treated with the same inhibitor. Breast cancer cells may have different genetic alterations not shared by glioblastoma cell lines such as the regulation of MMP2 and MMP9 expression. The authors should make a cautionary note.

RESPONSE: Concerning item 5 of your previous review, your question was concerning the use of MCF-7 and HT1080 in the MMP expression assays. These cell lines were used to provide positive controls for the activities of MMP2 and MMP9 in the zymogram assay, against which we could compare the glioma cells. This was particularly important to show that MMP9 was detected in the positive control but absent from the glioma cells. No treatments were applied to MCF-7 or HT1080, nor was there any comparison attempting to define the role of these inhibitors in these two cell lines. 

Point 2: However, a recent paper linking MMP2/9 with 15-lipoxygenase/15-HETE mediated via hypoxia activation may be of interest to the authors. https://www.sciencedirect.com/science/article/pii/S0022282818303328?via%3Dihub
The study may suggest the different effects on the MMP2/9 levels in the treatment between Luteolin and NDGA in Fig. 7.

RESPONSE: Concerning the hypoxia-mediated activation, this is interesting; however, the cells cultivated in our paper were performed in conditions of normoxia (please refer to our methodology section). Moreover, the cells involved in the paper you have cited are pulmonary epithelial cells and the authors were exploring angiogenesis while considering cardiovascular diseases. Their study, although interesting, involves a different context altogether than that of glioblastomas potentially migrating/invading tissue in the central nervous system. Therefore, the most that we can do is speculate about the influence of hypoxic vs. normoxic conditions on this pathway. Since there is little information in the literature available exploring this topic, publishing our study establishes a foundation from which we, and others, could explore the role of hypoxia in this context. Thank you for the idea.

Point 3: There are likely other references in different contexts and the authors are suggested to do literature search.

RESPONSE: Thank you for your suggestion. If you refer to our references section, you will find a comprehensive list of articles that elaborate the different themes discussed in our paper. It is very possible that there are more articles that exist because of the volume and rate at which information is produced, but a comprehensive review exclusively dedicated to elaborating the role of the LOX pathways in different tissue types would certainly be useful for colleagues in our field. Thank you for the idea.